

# Two-particle self-consistent approach for broken symmetry phases

Lorenzo Del Re[1,2]

**1** Max-Planck-Institute for Solid State Research, 70569 Stuttgart, Germany
**2** Institut für Theoretische Physik und Astrophysik and Würzburg-Dresden Cluster
of Excellence ct.qmat, Universität Würzburg, 97074 Würzburg, Germany

lorenzo.re@uni-wuerzburg.de

## Abstract

Spontaneous symmetry breaking of interacting fermion systems constitutes a major challenge for many-body theory due to the proliferation of new independent scattering channels once absent or degenerate in the symmetric phase. One example is given by the ferro/antiferromagnetic broken symmetry phase (BSP) of the Hubbard model, where vertices in the spin-transverse and spin-longitudinal channels become independent with a consequent increase in the computational power for their calculation. We generalise the non-perturbative Two-Particle-Self-Consistent method (TPSC) to address broken SU(2) magnetic phases in the Hubbard model, offering an efficient approach that incorporates strong correlations. We show that in the BSP, the sum-rule enforcement of susceptibilities must be accompanied by a modified gap equation resulting in a renormalisation of the order parameter, vertex corrections and the preservation of the gapless feature of the Goldstone modes. We then apply the theory to the antiferromagnetic phase of the Hubbard model in the cubic lattice at half-filling. We compare our results of double occupancies and staggered magnetisation to the ones obtained using Diagrammatic Monte Carlo showing excellent quantitative agreement. We demonstrate how vertex corrections play a central role in lowering the Higgs resonance with respect to the quasi-particle excitation gap in the spin-longitudinal susceptibility, yielding a well visible Higgs-mode.

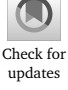

# 1   Introduction

The characterisation of broken-symmetry phases (BSP) in correlated quantum systems remains a formidable challenge for many-body theory. In fact, determining the precise ground state of spin Hamiltonians, such as the 3D-Heisenberg model with antiferromagnetic exchange, remains an open question to this day. Even though the precise ground state may remain elusive, it is possible to improve mean-field predicted groundstates, e.g. the Néel state, including quantum corrections encoded in the long-range and low-energy Goldstone modes [1–5], e.g. spin-waves in antiferromagnets [6]. The situation becomes richer when interacting electrons in solids are strongly correlated. A minimal model to describe such materials is the Hubbard model [7], where electrons interact through on-site Coulomb repulsion, enhancing electron localisation [8]. The theoretical challenge with strongly correlated BSP lies in simultaneously accounting for long-range fluctuations encoded in Goldstone modes and the localisation of electrons.

Such an ambitious task could be achieved by employing cluster [9] or diagrammatic [10] extensions of Dynamical Mean Field Theory (DMFT) [11], as well as Monte Carlo techniques [12–17]. However, the inclusion of long-range modes for cluster theories would be limited by the maximum size of the cluster used in the calculations, even if clever clustering schemes that permit an optimal finite-size scaling analysis are available [18]. In diagrammatic approaches, the proliferation of independent vertex components [19–25], once absent or degenerate in the symmetric phase, strongly increases the computational power needed for their numerical evaluation.

Hence, it is of great interest to develop efficient algorithms that require fewer computational resources while still accurately including correlation effects. In this context, the Two-Particle-Self-Consistent (TPSC) approach [26–33] has proven to be a reliable and efficient method for describing the physics of the Hubbard model in the weak-to-intermediate interaction regime. Given its reduced computational complexity, TPSC has already been successfully extended to multi-orbital models [30, 34], interfaced with *ab-intio* calculations [31] and applied to non-equilibrium [32]. However, current TPSC formulations are limited to symmetric phases, preventing their application to parameter regimes where materials exhibit broken symmetry phases (BSP). Additionally, because TPSC uses Moriya corrections to two-particle

propagator masses [35–39] to include correlation effects, a straightforward generalisation of TPSC equations might violate Goldstone's theorem, leading to an unphysical energy gap in the Goldstone modes. In this work, we extend the TPSC formalism to handle spontaneous symmetry breaking while correctly preserving the Goldstone modes.

We apply the new formulation to the antiferromagnetic phase of the three-dimensional Hubbard model on the cubic lattice. Our results show excellent quantitative agreement with Diagrammatic Monte Carlo (DiagMC) [16] across a wide range of interaction values. We demonstrate that as the temperature decreases from the critical value, the degree of correlation is reduced, which extends the theory's applicability to higher interaction values deep in the broken symmetry phase. Additionally, we show that symmetry breaking leads to a differentiation of vertex corrections in various scattering channels. This differentiation plays a central role in lowering the Higgs resonance relative to the quasi-particle excitation gap in the spin-longitudinal susceptibility, resulting in a clearly distinguishable Higgs mode.

The manuscript is organised as follows: in Sec. 2 we introduce the Hubbard model and establish the notation; Sec.3 describes the method and explains how two-particle self-consistency can be achieved in magnetic broken symmetry phases while preserving the Goldstone modes; in Sec.4 we show the numerical data of the order parameter and double occupancies comparing them with DiagMC, and we also show how TPSC is able to capture the elusive amplitude (Higgs) mode in the susceptibility spectra; in Sec.5 we provide our conclusions and outlook; in Appendix A we discuss some technical details relative to the derivation of the effective irreducible vertices; in Appendix B we present the derivation of the Bethe-Salpeter equations; in Appendix C we show the steps needed to obtain the corrected one-loop self-energy.

## 2 The model

In this work we will explicitly consider the single band Hubbard model in the cubic lattice,

$$H = -t \sum_{\langle ij \rangle \sigma} c_{i\sigma}^{\dagger} c_{j\sigma} + U \sum_{i} \hat{n}_{i\uparrow} \hat{n}_{i\downarrow}, \tag{1}$$

where $t$ is the electronic hopping amplitude between nearest-neighbours and $U$ is the local Coulomb repulsion. In the case of the AF phase, the system loses the full translational symmetry of the original cubic lattice and it is useful to introduce the sub-lattice index $a = A, B$ for specifying whether the fermionic field $c_{ia\sigma}^{\dagger}$ is evaluated at one site belonging to the sub-lattice A or B. Therefore, it is useful to introduce the generalised multi-flavor indices $\alpha$, which for example coincide with $\alpha = (a, \sigma)$ containing both sub-lattice ($a$) and spin ($\sigma$) indices in the AF or to spin indices in the FM case. Then, we can rewrite the Hubbard Hamiltonian in the following form:

$$H = \sum_{\langle ij \rangle} \sum_{\alpha\beta} c_{i\alpha}^{\dagger} \mathcal{H}^{\alpha\beta} c_{j\beta} + \frac{1}{2} \sum_{i} \sum_{\alpha\beta} U_{\alpha\beta} \hat{n}_{i\alpha} \hat{n}_{i\beta}. \tag{2}$$

In the case of FM, we have that $\mathcal{H}^{\alpha\beta} = -t\delta_{\alpha\beta}$ and $U_{\alpha\beta} = \delta_{\alpha\bar{\beta}} U$, whereas for the AF case we have $\mathcal{H}^{\alpha\beta} = -t\delta_{\sigma\sigma'}\delta_{a\bar{b}}$ and $U_{\alpha\beta} = \delta_{\sigma\bar{\sigma}'}\delta_{ab}U$, where $\bar{\ell}$ denote the opposite of index $\ell$, referring to the complementary spin or sub-lattice index (e.g., if $\ell$ is spin-up or sub-lattice A, then $\bar{\ell}$ is spin-down or sub-lattice B).

# 3 The method

The Two-Particle Self-Consistent (TPSC) method requires relatively low computational power and achieves its efficiency through a series of approximations, which we will examine in detail in this section. In practice, the self-energy is approximated in a form similar to that used in Hartree-Fock (HF) (see Figure 1). This assumption simplifies the expressions for the Green's function and two-particle susceptibilities, which can then be analytically obtained using a formula akin to the Random Phase Approximation (RPA).

However, unlike in HF and RPA, the vertex in the self-energy and susceptibility diagrams is represented not by the bare interaction but by an effective vertex. This effective vertex includes a renormalisation factor that depends on the double occupations, which are determined by imposing an exact sum rule on the susceptibility in the spin-transverse channel. This sum rule complements the 'usual' gap equation for the order parameter by coupling it to the double occupancies, which, unlike in HF, are determined self-consistently.

While we will provide an explicit derivation of all the equations, readers primarily interested in the results may refer to Figure 2, which presents a flow diagram summarizing the main steps and equations of the TPSC method.

## 3.1 The TPSC Ansatz

The core of TPSC consists in finding an approximate form for the electron self-energy from which one can construct a conserving approximation in the Baym-Kadanoff sense [40, 41]. In order to do so, one can start from the equation of motion that reads:

$$\Sigma^{\alpha\gamma}(x, y')G^{\gamma,\beta}(y', y) = U_{\alpha\gamma}G^{(2)\beta\alpha}_{\gamma\gamma}(y, x + 0^-, x + 0^+, x), \qquad (3)$$

where $G^{\alpha\beta}(x, y) = -T_t \left\langle c_\alpha(x)c_\beta^\dagger(y) \right\rangle$ is the Green's function, with $x = (R_i, \tau_i)$ being a four-vector containing the lattice coordinate $R_i$ and the imaginary time $\tau_i$, $c_\alpha(x) = e^{H\tau_i}c_{i\alpha}e^{-H\tau_i}$, $\Sigma^{\alpha\beta}(x, y)$ is the electronic self-energy, and

$$G^{(2)\alpha\beta}_{\gamma\delta}(x_1, x_2, x_3, x_4) = T_\tau \left\langle c_\alpha^\dagger(x_1)c_\beta(x_2)c_\gamma^\dagger(x_3)c_\delta(x_4) \right\rangle,$$

represents the two-particle Green's function. In Eq.(3), a summation is intended for the repeated indices $\gamma$ and $y'$. Due to the presence of $G^{(2)}$, Eq.(3) is not closed for the self-energy and single-particle Green's function, and in order to obtain an explicit expression for $\Sigma$ further approximations must be carried on. In mean-field theory the two-particle Green's function is replaced by its disconnected part, that is a valid approximation only at weak coupling. In TPSC [26, 27, 29, 30], in order to take into account of correlation effects, and at the same time to reduce the complexity of Eq.(3), the following assumption is considered:

$$\Sigma^{\alpha\gamma}(x, y')G^{\gamma,\beta}(y', y) \sim \lambda^{\alpha\gamma} U_{\alpha\gamma} \left[ G^{\alpha\beta}(x, y)n_\gamma - s^{\alpha\gamma}G^{\gamma\beta}(x, y) \right], \qquad (4)$$

where $n_\alpha = \langle \hat{n}_{i\alpha} \rangle$, $s^{\alpha\beta} = \left\langle c_{i\beta}^\dagger c_{i\alpha} \right\rangle$, and $\lambda^{\alpha\beta}$ is an extra-coefficient that must be determined self-consistently and contains correlation effects. When $\lambda^{\alpha\beta} = 1$ mean-field theory is recovered. The parameter $\lambda$ can be determined by requiring that the equal-time/position limit of Eq.(3), i.e. $y = x + 0^{++}$, is preserved exactly when $\beta = \alpha$,[1] by imposing:

$$\lambda^{\alpha\gamma} = \frac{\left\langle \hat{n}_\alpha \hat{n}_\gamma \right\rangle}{n_\alpha n_\gamma - s^{\alpha\gamma}s^{\alpha\gamma}}. \qquad (5)$$

---

[1]In the case of the AF phase that we address in this work, spin conservation implies that $\left\langle c_\alpha^\dagger \hat{n}_\gamma c_\beta \right\rangle = 0$ at zero field, when $\alpha \neq \beta$, and therefore we shall introduce the $\lambda$-correction only for the two-particle Green's functions that do not vanish in the limit of zero external field.

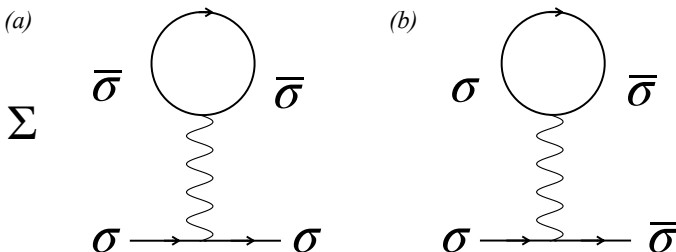

Figure 1: Diagrammatic representation of the diagonal (a) and off-diagonal (b) components of the self-energy, as analytically expressed in Eq.(8). The wiggly line represents the effective vertex $\Gamma_{\overline{\uparrow\downarrow}}$, which must be obtained self-consistently along with the order parameter (encoded in the Green's function, represented by the thick continuous line). Both are determined through the simultaneous solution of Eqs.(13,14). The diagrams shown here refer to the first-level self-energy, however in TPSC it is possible to calculate an improved version of $\Sigma$ which includes non-local and dynamical quantum corrections (see Section 3.2 and Figure 2).

From Eq.(4), we can isolate the self-energy that reads:

$$\Sigma^{\alpha\beta}(x-y) = \delta(x-y)\left(\delta_{\alpha\beta}\lambda^{\alpha\gamma}U_{\alpha\gamma}n_\gamma - \lambda^{\alpha\beta}U_{\alpha\beta}s^{\alpha\beta}\right). \tag{6}$$

In the FM/AF phase of the Hubbard model, the expression for the $\lambda$ parameter simplifies as follows:

$$\lambda = \frac{\langle \hat{n}_\uparrow \hat{n}_\downarrow \rangle}{n_\uparrow n_\downarrow}, \tag{7}$$

which is identical to the one in the paramagnetic case [27, 29]. Since in the FM/AF phase Eq.(7) does not depend on the spin/sub-lattice indices, we omitted those indices in the expression of $\lambda$, and therefore we need to optimize only one parameter even within the broken-symmetry phases under scrutiny. Hence, in the AF phase the expression of the self-energy can be written as:

$$\Sigma^{ab}_{\sigma\sigma'}(x,y) = \delta(x-y)\delta_{ab}U_{\overline{\uparrow\downarrow}}\left(\delta_{\sigma\sigma'}n_{a\bar{\sigma}} - \delta_{\sigma\bar{\sigma}'}s^{\sigma\bar{\sigma}}_a\right), \tag{8}$$

where $U_{\overline{\uparrow\downarrow}} = \lambda U$, $a, b$ are sub-lattice indices. We notice that in a AF phase the components of the self-energy off-diagonal in the spin indices should vanish, i.e. $s^{\sigma\bar{\sigma}} = 0$. However, it is useful to keep those terms in the expression of the self-energy for the derivation of the Bethe-Salpeter equation in the spin-transverse channel. Therefore, we will consider the presence of an external field that breaks spin-conservation and eventually compute the functional derivatives of $\Sigma$ with respect to the off-diagonal component of the propagator in the limit of a vanishing field.

In order to obtain self-consistency at the two-particle level, we have to calculate physical susceptibilities and therefore we need the knowledge of the irreducible vertex function $\Gamma$, which is obtained by carrying the functional derivative of $\Sigma$ with respect to $G$, i.e. $\Gamma(1,2,3,4) = \frac{\delta\Sigma(2,1)}{\delta G(3,4)}$ [42]. In the FM/AF phases the original SU(2) symmetry of the Hubbard Hamiltonian is spontaneously broken and the two independent scattering channels to be considered are the spin-transverse and spin-longitudinal channels [22].

### 3.1.1 Spin-transverse channel

The vertex function in the spin-transverse channel is defined as:

$$\Gamma^{abcd}_{\overline{\uparrow\downarrow}}(x_1,x_2,x_3,x_4) = \frac{\delta\Sigma^{ba}_{\downarrow\uparrow}(x_2,x_1)}{\delta G^{cd}_{\downarrow\uparrow}(x_3,x_4)} = -\lambda U\delta_{ab}\delta_{ac}\delta_{ad}\delta(x_1-x_2)\delta(x_1-x_3)\delta(x_1-x_4), \tag{9}$$

where we used Eq.(8) and the fact that $s_a^{\sigma\bar{\sigma}} = G_{\sigma\bar{\sigma}}^{aa}(x, x + 0^-)$.[2]

Let us now define the physical susceptibility in the spin-transverse channel:

$$\chi_{\sigma\bar{\sigma}}^{ab}(x_1, x_2) = T_\tau \left\langle S_a^{\sigma\bar{\sigma}}(x_1) S_b^{\bar{\sigma}\sigma}(x_2) \right\rangle, \tag{10}$$

where $S_a^{\sigma\sigma'}(x) = e^{H\tau} c_{ia\sigma}^\dagger c_{ia\sigma'} e^{-H\tau}$, with $x = (R_i, \tau)$. Since the vertex function in Eq.(9) is local and static, the Bethe-Salpeter equation (BSE) [see Appendix B for the derivation] for the physical susceptibilities is similar to the one obtained in RPA [22] and reads:

$$\bar{\bar{\chi}}_{\sigma\bar{\sigma}}^{-1}(q) = \bar{\bar{\chi}}_{0,\sigma\bar{\sigma}}^{-1}(q) + \bar{\bar{\Gamma}}_{\sigma\bar{\sigma}}, \tag{11}$$

where we used the double bar to indicate 2×2 matrices, $q = (i\omega_n, \mathbf{q})$ with $\omega_n = 2\pi n/\beta$ and $\mathbf{q}$ being respectively the bosonic Matsubara frequency and crystalline exchanged momentum, $\bar{\bar{\chi}}_{\sigma\bar{\sigma}}(q)$ is given by the Fourier transform of the susceptibility defined in Eq.(10), $\bar{\bar{\Gamma}}_{\sigma\bar{\sigma}} = -\lambda U \mathbb{I}_{2\times 2}$ and $\chi_{0,\sigma\bar{\sigma}}^{ab} = -\frac{1}{V\beta} \sum_k G_\sigma^{ab}(k) G_{\bar{\sigma}}^{ab}(k+q)$. The Green's function is obtained using the Dyson equation and reads:

$$\bar{\bar{G}}_\sigma^{-1}(k) = \epsilon_{\mathbf{k}} \sigma^x + [i\nu + \mu - \frac{\Gamma_{\overline{\uparrow\downarrow}}}{2}(n + \sigma m)]\mathbb{I}_{2\times 2}, \tag{12}$$

where $n$ is the electron density and $m = n_{A\uparrow} - n_{A\downarrow}$ is the staggered magnetisation.

In order to univocally determine single-particle and two-particle properties, we have to solve a set of self-consistent equations that will allow us to find the chemical potential, staggered magnetisation and double occupancies $(\mu, m, \langle \hat{n}_\uparrow \hat{n}_\downarrow \rangle)$ as a function of the electron density, on-site interaction and temperature. In this work we will specialize in the case of the three-dimensional cubic lattice at half-filling, i.e. $n = 1$, that corresponds to fixing the chemical potential to $\mu = \frac{|\Gamma_{\overline{\uparrow\downarrow}}|}{2}$.

Since the self-energy is static and local, the gap equation for the order parameter is similar to the one obtained in mean-field theory and is given by following expression:

$$\frac{1}{(2\pi)^3} \int_{BZ} d\mathbf{k} \frac{|\Gamma_{\overline{\uparrow\downarrow}}|}{2E_{\mathbf{k}}} \tanh\left(\frac{\beta E_{\mathbf{k}}}{2}\right) = 1, \tag{13}$$

where $E_{\mathbf{k}} = \sqrt{\epsilon_{\mathbf{k}}^2 + \left(\frac{m\Gamma_{\overline{\uparrow\downarrow}}}{2}\right)^2}$, with $\epsilon_{\mathbf{k}} = -2t\left[\cos(k_x) + \cos(k_y) + \cos(k_z)\right]$, which is obtained by imposing $m = \frac{1}{V\beta} \sum_{k\nu} e^{i0^-\nu}[G_\uparrow^{AA}(k) - G_\downarrow^{AA}(k)]$ and by substituting Eq.(12) into the last equation. Differently from mean-field theory however, the order parameter is not univocally determined by the gap equation, because the double occupancies, appearing in Eq.(13), are still unknown.

As a direct consequence of its definition in Eq.(10), the susceptibility in the transverse channel assumes the following limiting value $\sum_\sigma \chi_{\sigma\bar{\sigma}}^{aa}(x, x + 0^-) = n - 2\langle \hat{n}_\uparrow \hat{n}_\downarrow \rangle$, which implies the following sum rule for its Fourier transform:

$$\frac{1}{\beta(2\pi)^3} \sum_{\omega_n \sigma} \int_{BZ} d\mathbf{q}\, \chi_{\sigma\bar{\sigma}}^{aa}(q) = n - 2\langle \hat{n}_\uparrow \hat{n}_\downarrow \rangle. \tag{14}$$

Hence, Eqs.(13,14) provide with a closed set of equations that must be solved self-consistently in order to determine the order parameter and the double occupancies.

---

[2] We used the overline symbol, i.e. $\overline{\uparrow\downarrow}$, to distinguish this vertex component from those belonging to the spin-longitudinal channel, that are defined in the next paragraphs.

## 3.2 Spin-longitudinal channel

The irreducible vertex function in the spin-longitudinal channel reads:

$$\Gamma_{\sigma\sigma'}^{abcd}(x_1, x_2, x_3, x_4) = \frac{\delta\Sigma_{\sigma\sigma}^{ba}(x_2, x_1)}{\delta G_{\sigma'\sigma'}^{cd}(x_3, x_4)} \sim U_{\sigma\sigma'}\delta_{\sigma\bar{\sigma}'}\delta_{ab}\delta_{ac}\delta_{ad}\,\delta(x_1-x_2)\delta(x_1-x_3)\delta(x_1-x_4). \quad (15)$$

Differently from Eq.(9) which is an exact equality, a further approximation, similar to the one performed in the charge channel in the paramagnetic phase [26,29], is needed to write Eq.(15) in its final form (see Appendix A).

Let us define the susceptibilities in the spin-longitudinal channel:

$$\chi_{\sigma\sigma'}^{ab}(x_1, x_2) = T_\tau \langle n_{a\sigma}(x_1)n_{b\sigma'}(x_2)\rangle - \langle n_{a\sigma}\rangle\langle n_{b\sigma'}\rangle. \quad (16)$$

Given the local and static form of the vertex function in Eq.(15), the expression of the susceptibilities in the charge and spin-longitudinal channel can be written as follows:

$$\bar{\bar{\chi}}_\parallel^{-1}(q) = \bar{\bar{\chi}}_{0,\parallel}^{-1}(q) + \bar{\bar{\Gamma}}_\parallel, \quad (17)$$

where:

$$\bar{\bar{\chi}}_{0,\parallel}(q) = \begin{pmatrix} \chi_{0,\uparrow\uparrow}^{AA} & \chi_0^{AB} \\ \chi_0^{AB} & \chi_{0,\downarrow\downarrow}^{AA} \end{pmatrix}, \quad (18)$$

$$\bar{\bar{\Gamma}}_\parallel(q) = \begin{pmatrix} \Gamma_{\uparrow\uparrow} & \Gamma_{\uparrow\downarrow} \\ \Gamma_{\downarrow\uparrow} & \Gamma_{\downarrow\downarrow} \end{pmatrix}, \quad (19)$$

with $\chi_{0,\sigma\sigma}^{ab}(q) = -\frac{1}{V\beta}\sum_k G_\sigma^{ab}(k)G_\sigma^{ab}(k+q)$.

In general, the vertex appearing in Eq.(19) must be determined self-consistently by imposing the correct sum-rule for each vertex component. There are three independent sum rules for the longitudinal channel that read:

$$\frac{2}{\beta(2\pi)^3}\sum_{\omega_n}\int_{\text{BZ}} d\mathbf{q}\,\chi_z(q) = n - 2\langle\hat{n}_\uparrow\hat{n}_\downarrow\rangle - m^2, \quad (20)$$

$$\frac{2}{\beta(2\pi)^3}\sum_{\omega_n}\int_{\text{BZ}} d\mathbf{q}\,\chi_\rho(q) = n + 2\langle\hat{n}_\uparrow\hat{n}_\downarrow\rangle - n^2, \quad (21)$$

$$\frac{2}{\beta(2\pi)^3}\sum_{\omega_n}\int_{\text{BZ}} d\mathbf{q}\,\chi_{z\rho}(q) = m(1-n), \quad (22)$$

where we expressed the susceptibility and vertex components in the physical basis as: $\chi_z = \frac{1}{4}\sum_{ab\sigma\sigma'}(-1)^{a+b+\sigma+\sigma'}\chi_{\sigma\sigma'}^{ab}$, $\chi_\rho = \frac{1}{4}\sum_{ab\sigma\sigma'}\chi_{\sigma\sigma'}^{ab}$, $\chi_{z\rho} = \frac{1}{4}\sum_{ab\sigma\sigma'}(-1)^{a+\sigma}\chi_{\sigma\sigma'}^{ab}$, $\Gamma_z = -\frac{1}{2}\sum_{\sigma\sigma'}(-1)^{\sigma+\sigma'}\Gamma_{\sigma\sigma'}$, $\Gamma_\rho = \frac{1}{2}\sum_{\sigma\sigma'}\Gamma_{\sigma\sigma'}$, $\Gamma_{z\rho} = \frac{1}{2}\sum_{\sigma\sigma'}(-1)^\sigma\Gamma_{\sigma\sigma'}$. For generic fillings, the mixed components $\chi_{z\rho}$ and $\Gamma_{z\rho}$ are nonzero and must be determined [43]. However, at half-filling $\chi_{z\rho}$ and $\Gamma_{z\rho}$ vanish exactly [22] and with them Eq.(22). Therefore, in this case the charge and spin-longitudinal channels are decoupled and their susceptibilities can be written as:

$$\chi_z(q) = \frac{\chi_{0,\parallel}(q)}{1 - \Gamma_z\chi_{0,\parallel}(q)}, \quad (23)$$

$$\chi_\rho(q) = \frac{\chi_{0,\parallel}(q)}{1 + \Gamma_\rho\chi_{0,\parallel}(q)}, \quad (24)$$

where $\chi_{0,\parallel} = -\frac{1}{2V\beta}\sum_{k\sigma b} G_\sigma^{Ab}(k)G_\sigma^{bA}(k+q)$.

Since Eqs.(13,14) are a set of closed equations, Eqs.(20,21) can be solved separately once the values of $m$ and $\langle\hat{n}_\uparrow\hat{n}_\downarrow\rangle$ have been self-consistently obtained from the spin-transverse channel.

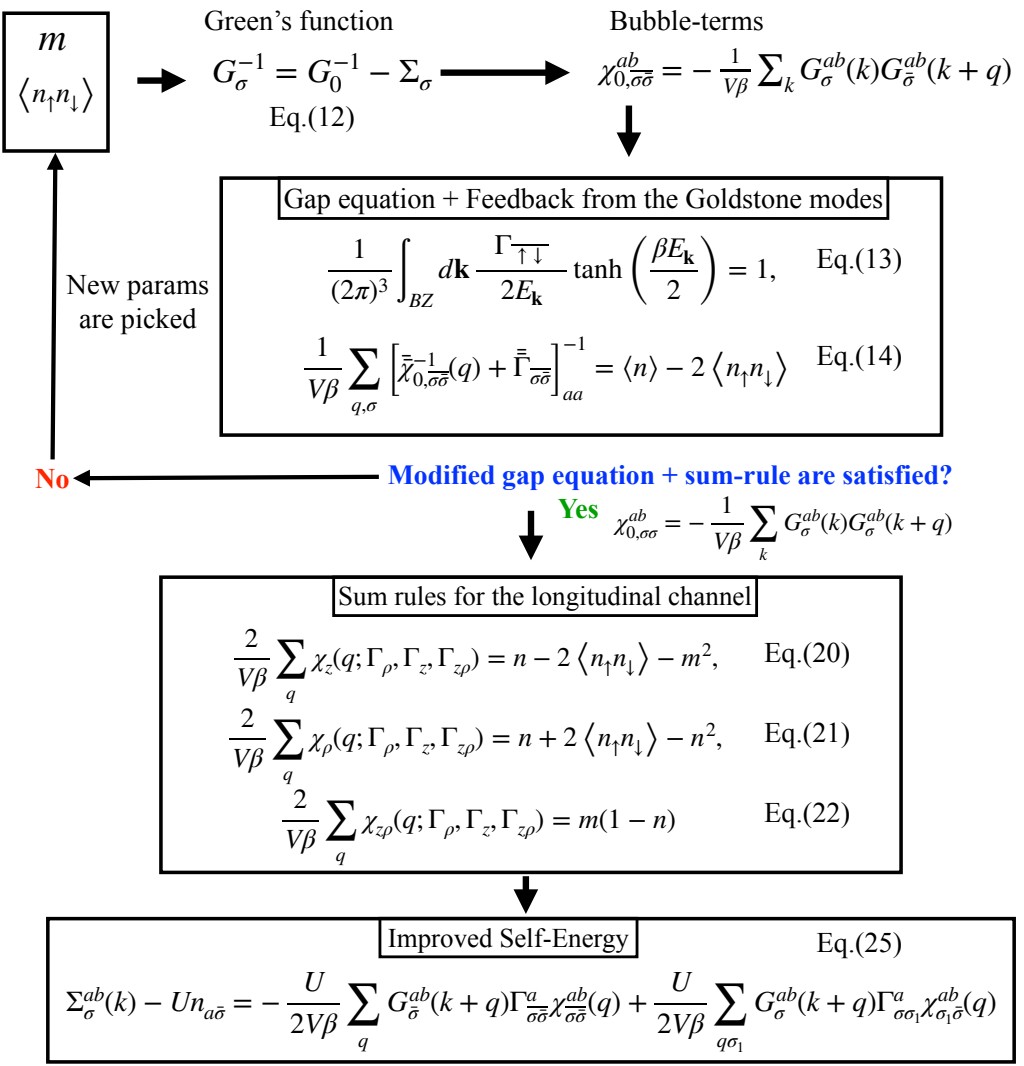

Figure 2: **Work-flow of the Two-Particle Self-Consistent (TPSC) method for anti-ferromagnetic phases of the Hubbard model.** The first box at the top shows how the staggered magnetisation $m = n_{A\uparrow} - n_{A\downarrow}$ and double occupancies are obtained self-consistently by solving the gap equation [Eq. (13)] and the sum rule [Eq. (14)] for the spin-transverse channel, where Goldstone modes appear. An initial guess for $m$ and $\langle n_\uparrow n_\downarrow \rangle$ is used to calculate the Green's function and susceptibility. If Eqs. (13) and (14) are not satisfied, a minimisation routine adjusts the values. Once satisfied, the next step is to find the renormalised vertices in the spin-longitudinal channel by enforcing Eqs. (20), (21),and (22) shown in the middle box. With all renormalised interactions in the different channels, the improved electron self-energy can finally be computed using Eq. (25) displayed in the box at the bottom.

## 3.3 Improved one-loop self-energy

In TPSC it is possible to obtain an improved self-energy that, differently from the one appearing in Eq.(8), depends on both momenta and frequency. This can be achieved by computing the TPSC vertices and susceptbilities and using them as input for the equation of motion [27, 44]. Extending this procedure to the broken symmetry phase we obtain the following expression

for the improved self-energy:

$$\Sigma_\sigma^{ab}(k) - U n_{a\bar{\sigma}} = -\frac{U}{2V\beta}\sum_q G_{\bar{\sigma}}^{ab}(k+q)\Gamma_{\sigma\bar{\sigma}}^a \chi_{\sigma\bar{\sigma}}^{ab}(q) + \frac{U}{2V\beta}\sum_{q\sigma_1} G_\sigma^{ab}(k+q)\Gamma_{\sigma\sigma_1}^a \chi_{\sigma_1\bar{\sigma}}^{ab}(q), \quad (25)$$

where $G_\sigma^{ab}(k)$ is given by Eq.(12). In appendix C we show the derivation of Eq.(25).

## 4 Numerical results

Fig. (3-a) shows the order parameter as a function of temperature for different values of the on-site interaction. The order parameter decreases as a function of increasing temperature until it vanishes at the critical temperature. Close to the phase transition, the order parameter behaves like $m = \alpha|T - T_c|^\beta$ with critical exponent $\beta = 1/2$, which is different from the exact one belonging to the $O(3)$ (Heisenberg) universality class $\beta \sim 0.369$ [45]. The exponent value for the order parameter $\beta = \frac{1}{2}$ might suggest that TPSC is a mean-field theory. However, the method actually belongs to a different universality class. The critical exponents for TPSC, as well as those for other theories based on $\lambda$-Moriya [35, 36] corrections [see, for example, [37, 38, 46], fall within the $O(N)$ universality class in the limit $N \to \infty$ [28]. This class is distinct from the mean-field, which is obtained in the limit of infinite spatial dimensions and is characterized by the exponents $\nu = \frac{1}{2}$, $\gamma = 1$, and $\beta = \frac{1}{2}$. In contrast, in three dimensions, the critical exponents for $O(\infty)$ are $\nu = 1$, $\gamma = 2$, and $\beta = \frac{1}{2}$. In two dimensions, the critical temperature vanishes, as predicted by the Mermin-Wagner theorem, which also holds for the $O(3)$ case. This can be understood by considering the divergence of the sum in Eq.(14) at finite temperature in 2D within the broken symmetry phase, while it remains finite at $T = 0$, where the discrete sum over Matsubara frequencies is replaced by an integral over a continuous variable. Our results for the critical exponent $\beta = \frac{1}{2}$ is therefore consistent with previous calculations showing that TPSC belongs to the $O(\infty)$ universality class.

In Fig.(3-b), we show the value of the vertex renormalisation $\lambda = |\Gamma_{\overline{\uparrow\downarrow}}|/U$ as a function of temperature for different values of $U$. We observe that $\lambda$ decreases as a function of increasing interactions, as expected, since the system get more correlated when $U$ increases. On the other hand, $\lambda$ increases by decreasing the temperature from the critical one, which can be rationalised in the following way: when symmetry breaking is allowed, the system can reduce the number of double occupancies $\langle \hat{n}_\uparrow \hat{n}_\downarrow \rangle = \frac{\lambda}{4}(n^2 - m^2)$, shown in Fig.(3-c), (and therefore minimize the potential energy) by increasing the order parameter, rather than by decreasing $\lambda$. Hence, our results show that the degree of correlation of the system is reduced deep in the broken symmetry phase far away from the the critical temperature.

In Fig.(3-d), we show the order parameter and double occupancies as a function of $U$ by fixing the temperature to $T/t = 1/10$. As expected we observe that the order parameter (double occupancies) increases (decrease) as a function of $U$. It is worth to highlight that the introduction of quantum fluctuations leads to a significant decrease in the staggered magnetisation compared to its mean-field predicted value [black curve in Fig. (3-d)]. We compared our results to the ones obtained using Monte Carlo in Ref. [16] and we observe an excellent quantitative agreement. It is worth noting that HF deviates significantly from the exact DiagMC results and the TPSC ones even at weak coupling. This behaviour is somewhat similar to what has been predicted in the symmetric phase near criticality, where second-order perturbation theory shows a sizeable deviation from mean-field predictions [47].

After solving Eqs.(13,14) we can use the values of double occupations and staggered magnetisation as input for Eqs.(20,21) in order to obtain the renormalised vertices in the longitudinal channel. In Fig.(4), we show the renormalisation factors of the vertices, i.e. $\Gamma_\rho/U$, $\Gamma_z/U$ and $\lambda$ as a function of $U$ for $T/t = 1/7.5$. We observe that $\Gamma_\rho$ is highly enhanced with respect

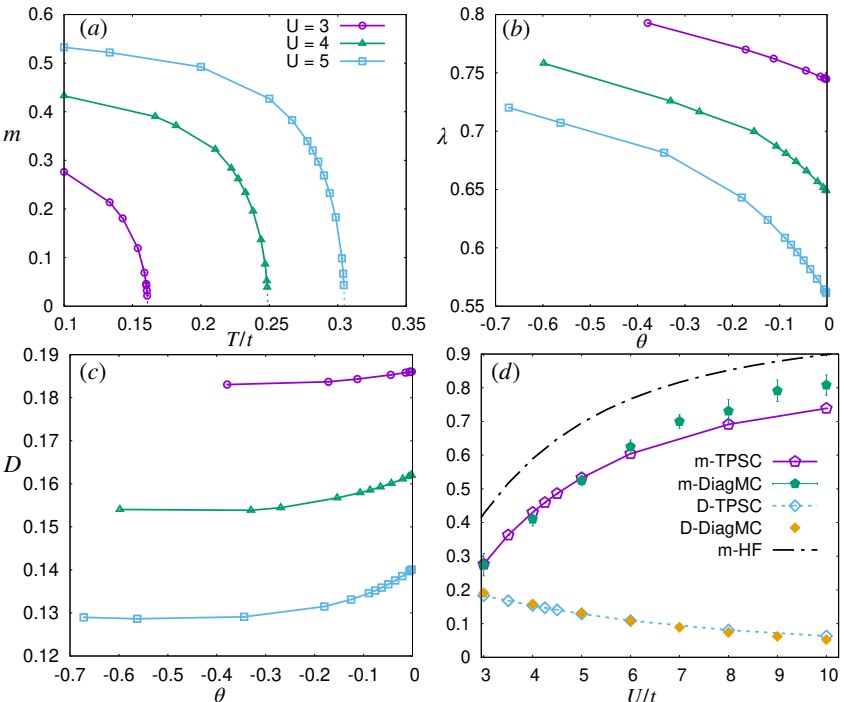

Figure 3: (a) Staggered magnetisation $m$ as a function of $T$ for three different values of $U/t = 3, 4, 5$. Dashed lines are best fits of the function $\alpha |T - T_c|^{1/2}$ close to $T_c$. (b) $\lambda$ parameter as a function of the reduced temperature $\theta = \frac{T - T_c}{T_c}$ for the three different values of the on-site interaction. (c) Double occupancies $D = \left\langle \hat{n}_\uparrow \hat{n}_\downarrow \right\rangle$ as a function of $\theta$ for the three different $U$ values. (d) Magnetisation and double occupancies as a function of $U$ for $T/t = 1/10$. TPSC data (open symbols) are compared to the DiagMC results (filled symbols) adapted from Ref. [16]. The black dashed line is the magnetisation curve obtained using Hartree-Fock.

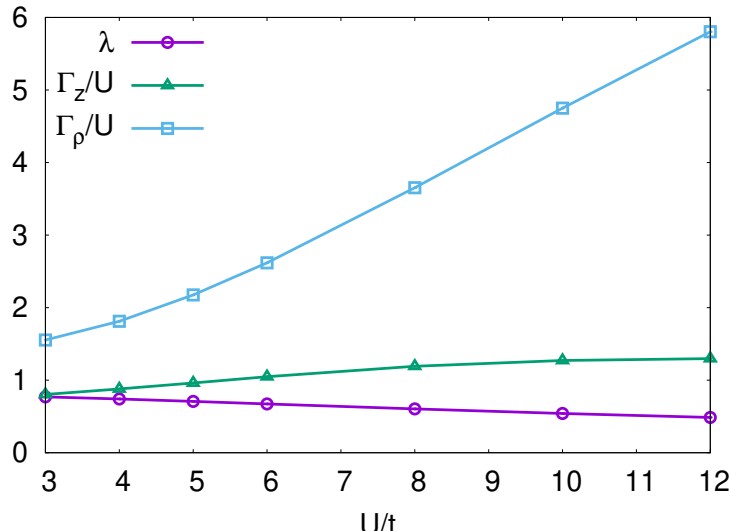

Figure 4: Vertex renormalisations in the density ($\Gamma_\rho/U$), spin-longitudinal ($\Gamma_z/U$) and spin transverse ($\lambda = |\Gamma_{\overline{\uparrow\downarrow}}|/U$) channels as a function of the bare interaction for $T/t = 1/7.5$.

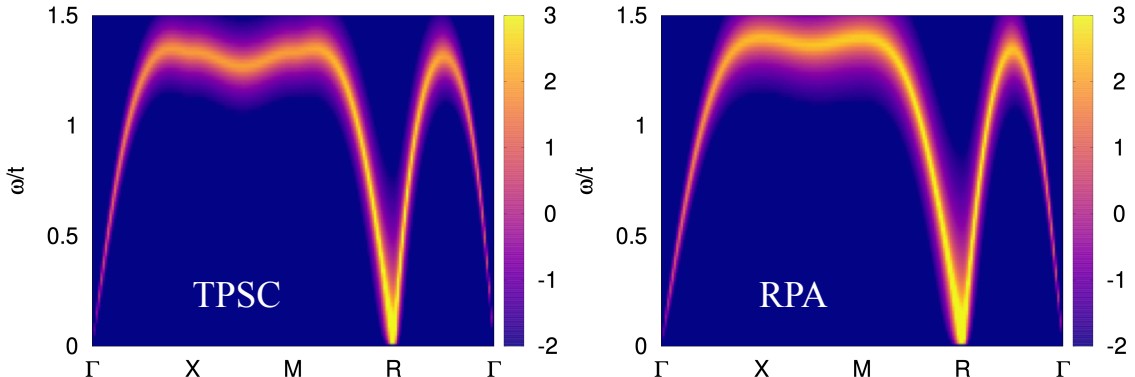

Figure 5: Imaginary part of the spin-transverse susceptibilities $\chi_x(\omega + i\eta)$ for $U/t = 5$, $T/t = 1/7.5$ , $\eta = 0.03$ calculated using TPSC (left panel) and RPA (right panel).

to the bare vertex which is similar to what has been already observed in the paramagnetic phase of the Hubbard model using TPSC [27]. Differently from the symmetric case, in the AF phase $\Gamma_z \neq |\Gamma_{\overline{\uparrow\downarrow}}|$, and our results show that $\Gamma_z > |\Gamma_{\overline{\uparrow\downarrow}}|$ for all values of $U$ and that the difference between the two vertices increases as a function of the on-site interaction. Interestingly, while $|\Gamma_{\overline{\uparrow\downarrow}}|$ is always lower than the bare vertex (as $U_s$ in the paramagnetic phase [27]), this is not true anymore for $\Gamma_z/U$, which is also an increasing function of $U$ and crosses the unity at $U/t \sim 5.4$ for $T/t = 1/7.5$ [see Figure 4].

Integrals in the Brillouin zone were numerically calculated using the trapezoidal rule in three dimensions, employing grids of $N_k \times N_k \times N_k$ points with $N_k$ values up to 32. For the numerical integration of the spin-transverse susceptibility evaluated at zero frequency, i.e., $\int d\mathbf{q} \sum_\sigma \chi_{\sigma\bar{\sigma}}(\mathbf{q}, 0)$, a specific strategy was applied. Since this function diverges at $\mathbf{q} = \mathbf{\Pi}$, that point was excluded from the integration grid. We evaluated the integral for different $N_k$ values (21, 24, 28, 32) and then extrapolated the integral value by fitting the function $I + h/N_k$, where $I$ represents the extrapolated value. For the summation over Matsubara frequencies, we evaluated the momentum integrals up to 24 bosonic frequencies. We then performed a fit to extrapolate the high-frequency quadratic tails, which allowed us to extend the summation to thousands of frequencies.

## 4.1 Dynamical susceptibilities

We can use the solution of the self-consistent equations to evaluate spectral properties of two-particle propagators. Regarding the spin-transverse channel, we observe that self-energy and vertex corrections are both controlled by the same quantity, i.e. $\Gamma_{\overline{\uparrow\downarrow}}$, which substitutes *de facto* the bare vertex appearing in RPA. Therefore, the spin-transverse dynamical susceptibility defined in Eq.(11), which contains the information about the Goldstone modes, calculated at a given $U$ corresponds to the RPA one evaluated at a lower value of the interaction, namely $|\Gamma_{\overline{\uparrow\downarrow}}(U)|$. In Figure 5, we show the imaginary part of the spin-transverse susceptibility $\mathrm{Im}\chi_x(\omega + i\eta, \mathbf{q})$ evaluated on the real axis, where $\chi_x = \frac{1}{2}(\chi_{\overline{\uparrow\downarrow}}^{AA} + \chi_{\overline{\uparrow\downarrow}}^{BB}) + \chi_{\overline{\uparrow\downarrow}}^{AB}$. The TPSC spectrum, much like that derived from RPA, accurately predicts the existence of Goldstone gapless modes at the R-point in the Brillouin zone. While TPSC introduces a quantitative renormalisation to the low-energy modes, the qualitative behaviour remains consistent with RPA.[3] Conversely, the vertex in the spin-longitudinal susceptibility $\Gamma_z$ assumes different values

---

[3]We evaluated the spectra on the real axis using the analytical expressions [22] for the bubble terms and used a grid of 32 × 32 × 32 internal momenta.



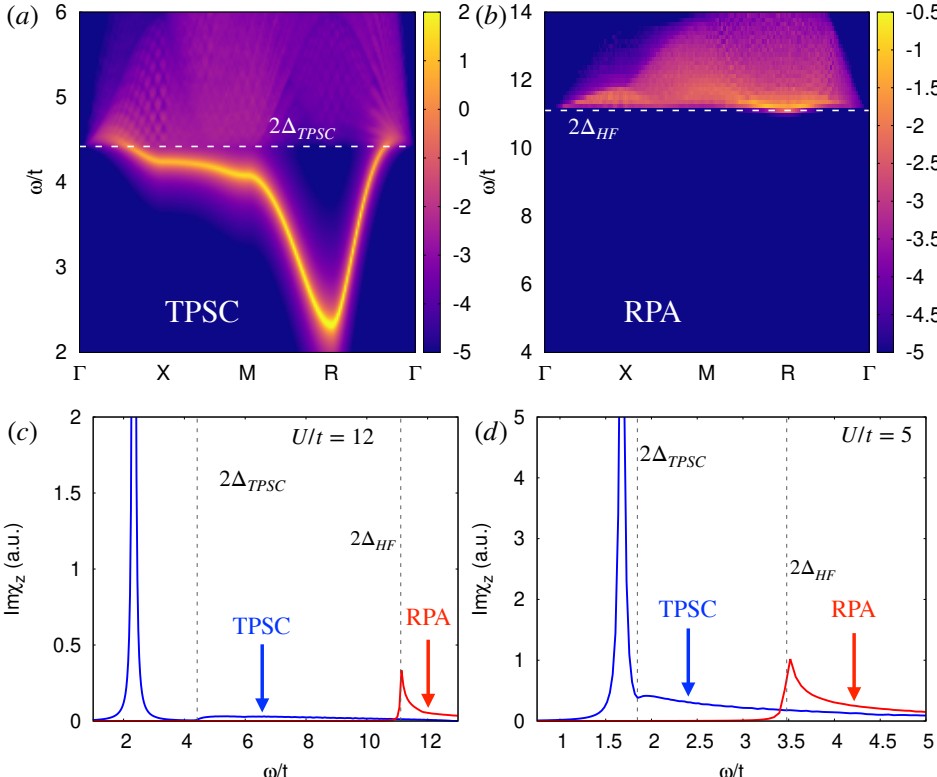

Figure 6: (a) Imaginary part of $\chi_z(\omega + i\eta, \mathbf{q})$ (in log scale) defined in Eq.(23) evaluated along the BZ high-symmetry path and for a wide range of real frequencies, for $U/t = 12$, $T/t = 1/7.5$, and $\eta/t = 0.03$. (b) Imaginary part of $\chi_z(\omega + i\eta, \mathbf{q})$ (in log scale) calculated using RPA for $U/t = 12$, $T/t = 1/7.5$, and $\eta/t = 0.02$. (c) $\mathrm{Im}\chi_z(\omega+i\eta, \mathbf{q})$ evaluated using TPSC and RPA at fixed momentum $\mathbf{q} = (\pi, \pi, \pi + 0.1)$ at $U/t = 12$, $T/t = 1/7.5$ and for $\eta = 0.03$. (d) Same as (c) but for $U/t = 5$.

than $\Gamma_{\uparrow\downarrow}$ because of symmetry breaking, and $\Gamma_z > |\Gamma_{\uparrow\downarrow}|$ as shown in Figure 4. This implies that the spin-longitudinal susceptibility evaluated in TPSC does not correspond to any RPA one evaluated at different effective parameters, and consequently the two methods yield qualitatively different results for the spin-longitudinal susceptibility. In particular, since $\Gamma_z > |\Gamma_{\uparrow\downarrow}|$ the gap in the $\chi_z$ spectrum is reduced with respect to the quasi-particle gap predicted by TPSC, i.e. $2\Delta_{\mathrm{TPSC}} = |\Gamma_{\uparrow\downarrow}| m$, which is controlled by self-energy corrections. In Fig.(6-a) we show a color plot of $\mathrm{Im}\chi_z(q)$ that has been evaluated in the high-symmetry path of the BZ and for a wide range of frequencies at $U/t = 12$ and $T/t = 1/7.5$. We observe that a well visible Higgs mode appears well below the quasi-particle continuum starting at $2\Delta_{\mathrm{TPSC}}$, it has a minimum at $R = (\pi, \pi, \pi)$, and presents a substantial dispersion along the M-R and R-$\Gamma$ directions. This is in stark contrast with the RPA predicted spectrum [shown in Fig.(6-b)], where the Higgs resonance occurs at $\omega/t = 2\Delta_{\mathrm{HF}}$ and therefore is overdamped by the particle-hole continuum [48, 49]. Our findings agree qualitatively with recent numerical results based on a time-dependent Gutzwiller approach showing that the Higgs resonance is shifted below the edge of the particle-hole continuum upon increasing the interaction [50]. In Figs.(6-c,d) we show $\mathrm{Im}\chi_z$ evaluated using TPSC and RPA as a function of the real frequencies for a fixed momentum close to $R$ and two values of the interactions $U/t = 12, 5$ and at $T/t = 1/7.5$. It is apparent that for both values of the interaction the Higgs resonance predicted by TPSC is well separated from the particle-hole continuum and occurs at lower energies, while RPA does not yield any true isolated pole.

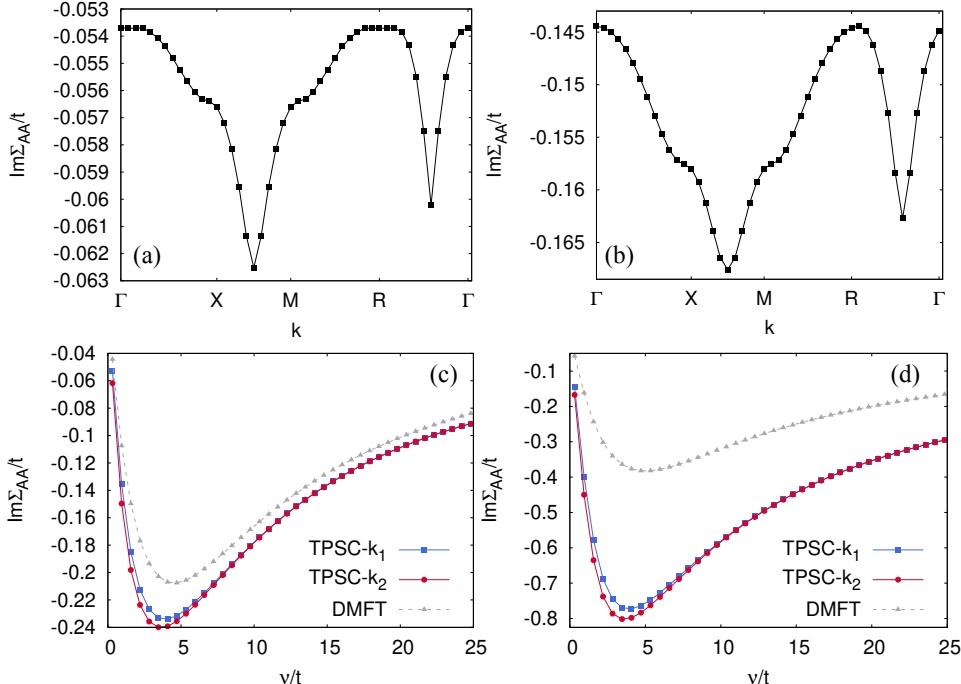

Figure 7: (a) Imaginary part of the self-energy as a function of the crystalline momentum for $\sigma = \uparrow$, $U/t = 3$, $T/t = 1/10$ evaluated at $\nu = \pi/\beta$. (b) Same as (a) but for $U/t = 5$. (c) Imaginary part of the self-energy as a function of the fermionic Matsubara frequency evaluated at two different points in the BZ $k_1 = (0,0,0)$ [blue squares] and $k_2 = (\pi, \frac{\pi}{2}, 0)$ [red circles] and for $U/t = 3$, $T/t = 1/10$. The imaginary part of the (local) self-energy evaluated in DMFT for the same parameters is represented by gray triangles. (d) Same as (c) but for $U/t = 5$.

## 4.2 Improved self-energy

In this section, we discuss numerical results for the improved self-energy obtained by incorporating TPSC collective modes into the equation of motion [Eq. (25)]. Unlike the mean-field-like self-energy shown in Figure 1, the improved self-energy exhibits frequency and momentum dependence. Figure (7a) displays the imaginary part of the electron self-energy for the majority spin species as a function of crystalline momentum, with $U/t = 3$ and $T/t = 1/10$, evaluated at the first Matsubara frequency $\nu = \pi/\beta$. We observe that Im$\Sigma$ peaks in absolute value at momenta $k = (\pi, \frac{\pi}{2}, 0)$ and $k = (\frac{\pi}{2}, \frac{\pi}{2}, \frac{\pi}{2})$, where the gap between the quasiparticle bands reaches its minimum value. Figure (7b) shows the same quantity for $U/t = 5$. Here, while the qualitative behaviour of the self-energy remains similar, its overall magnitude increases significantly.

We next examine the frequency dependence of the self-energy at fixed crystalline momentum. Figure (7c) illustrates Im$\Sigma$ as a function of Matsubara frequency for two chosen momenta: $k_1 = \Gamma$ (blue squares) and $k_2 = (\pi, \frac{\pi}{2}, 0)$ (red dots), where the self-energy reaches its extreme values at the lowest frequency, for $U/t = 3$, $T/t = 1/10$. For comparison, we include DMFT results from the antiferromagnetic solution (gray triangles) [51]. At $U/t = 3$, both methods show good agreement overall, though some noticeable differences emerge at higher frequencies. Despite the moderate $k$-dependence of the electron self-energy (variation of around 16% at the lowest frequency), the two methods display similar qualitative features. For $U/t = 5$ [Figure 7d], the quantitative deviation between TPSC and DMFT becomes more pronounced, with the TPSC self-energy having a greater magnitude. This dis-

crepancy is expected, as DMFT does not account for gapless quantum fluctuations from Goldstone modes due to its local, single-site formulation. Specifically, DMFT lacks two-particle self-consistency, meaning that the local spin fluctuations obtained from the effective Anderson impurity model (AIM) do not match the sum of the Fourier components of the lattice susceptibility: $\chi^{AIM}_{\uparrow\downarrow,loc}(\omega) \neq \frac{1}{V}\sum_{\mathbf{q}}\chi^{DMFT}_{\uparrow\downarrow}(\mathbf{q},\omega)$ [38]. Consequently, while $\chi^{DMFT}_{\uparrow\downarrow}(\mathbf{q},\omega)$ may correctly predict Goldstone modes [21], these modes do not influence the DMFT self-energy, which remains a purely local quantity. Although TPSC captures electron scattering with collective modes, the Green's function in Eq. (25) lacks self-energy damping. This limitation may lead to overestimated quantum corrections in TPSC. A comprehensive comparison with dynamical quantities calculated using DiagMC in the broken symmetry phase could further clarify TPSC's strengths and limitations, which we leave for future work. We expect that the applicability of TPSC is limited in regions of parameter space where the dynamical structure of the vertex function cannot be neglected [52–55].

## 5 Conclusions

We extended the formalism of TPSC to account for spontaneous symmetry breaking and applied the new method to the AF phase of the single-band Hubbard model on the cubic lattice at half-filling. Our comparison with DiagMC reveals excellent quantitative agreement between the two methods for the order parameter and double occupancies.

We show that the differentiation of vertex corrections in the different scattering channels due to symmetry breaking ($\Gamma_z \neq |\Gamma_{\overline{\uparrow\downarrow}}|$) has remarkable effects in the spin-longitudinal channel. In particular, the Higgs resonance occurs at energies lower than the quasi-particle continuum leading to a well visible Higgs mode for a wide range of parameters.

In TPSC, an improved electron self-energy can be constructed, exhibiting a nontrivial frequency and momentum structure, as shown in the latter part of our results section. Although we observe a limited dependence on the crystalline momentum, the TPSC self-energy is generally larger in magnitude compared to that obtained from DMFT, similarly to what is found in ladder-DΓA in the paramagnetic phase close to criticality [46]. This difference arises because TPSC incorporates Goldstone modes in the self-energy calculation, whereas DMFT does not. We leave to future work the exploration of doped antiferromagnetic states, where the momentum dependence of the electron self-energy could become more pronounced.

Since our data demonstrate that the level of correlation decreases by decreasing temperature deep in the BSP, one could argue that TPSC is particularly suited to the study of BSP where correlation are not negligible but less pronounced.

Additionally, TPSC has already been successfully integrated with ab-initio methods, though only for symmetric phases [31]. This opens up exciting possibilities for extending our method to broken symmetry phases in combination with DFT (Density Functional Theory) for realistic electronic structure calculations.

Also, since TPSC already has been used as a benchmark for cold atomic simulators [56,57], its generalisation will provide further guidance to cold-atom experiments exploring broken symmetry phases [58].

Generalising improved versions of TPSC, such as TPSC+ and TPSC+SFM [33], to the BSP case could lead to the partial inclusion of dynamical effects. These effects are particularly important near the Néel temperature [59,60], and will be addressed in future work.

Additionally, combining TPSC with DMFT [61,62] in the antiferromagnetic phase could provide deeper insights into the non-local quantum corrections to the spin-polaron peaks that emerge at strong coupling in the Heisenberg regime [51,63,64]. Furthermore, similar steps as those presented in this work could be applied to extend TPSC to study charge density waves

and superconductivity in the attractive and extended Hubbard models [46, 65–68].

The potential for applying TPSC to understand complex magnetic phases in novel materials is vast. For example, the approach we present here can be applied to models hosting altermagnetism [69–72], a recently identified category of broken-symmetry phases. Group theory predictions suggest that many such materials might exist in three dimensions [73], providing an ideal scenario where our method can be readily applied. Investigations of these novel magnetic phases in candidate compounds [74–79] are underway, and we anticipate that new magnetic materials will soon be proposed theoretically and realized experimentally. We also demonstrated that TPSC is an effective tool for studying the amplitude (Higgs) mode, which is often elusive in most mean-field theories. This paves the way for theoretical calculations of amplitude collective modes in altermagnets, providing a reference for future experimental investigations and offering insights into fundamental questions–such as how the topological properties of altermagnets electronic structures [80, 81] are reflected in their collective modes.

Let us note that in principle, the same scheme presented here can be applied to ordered states with larger unit cells, though the technical challenges depend on the type of incommensurate order. For spiral order, where the order parameter rotates in a plane with momentum $Q$ (e.g., $m_R \propto (\cos(Q \cdot R), \sin(Q \cdot R), 0)$), the computation is simplified by re-expressing the Hamiltonian in a new basis, restoring translational symmetry. This approach is similar to that used in studies of the Hubbard model with artificial gauge fields [82] or multi-orbital generalisations of coplanar magnetic states [83]. However, for striped collinear order, where the order parameter amplitude is modulated (e.g., $m_R \propto (0, 0, \cos(Q \cdot R))$), the enlarged unit cell must be explicitly considered [84], increasing computational cost due to the inclusion of additional orbitals.

# Acknowledgments

I thank Walter Metzner, Alessandro Toschi, Georg Rohringer, Thomas Schäfer and Lara Benfatto for valuable discussions. I also thank Renaud Garioud for providing the DiagMC data.

# A Irreducible vertices

In this section we shall give some details about the derivation of the expression for the irreducible vertices in the spin-transverse and spin-longitudinal channels.

## A.1 Spin-transverse channel

It is worth to note that the expression for the irreducible vertex in the spin-transverse channel presented in the main text is an exact equality. In fact, even if $\lambda$ is a functional of the Green's function, it does not appear in the expression of the irreducible vertex function because its functional derivative with respect to the off-diagonal propagator vanishes, i.e.

$$\frac{\delta \lambda}{\delta G_{\downarrow\uparrow}^{cd}(x_3, x_4)} = 0. \tag{A.1}$$

In fact, from Eq.(3) we can derive the following formula for the double occupancies:

$$\langle \hat{n}_{a\sigma} \hat{n}_{a\bar{\sigma}} \rangle = \frac{1}{2U} \Sigma_{\sigma\sigma'}^{aa'}(x, y') G_{\sigma'\sigma}^{a'a}(y', x). \tag{A.2}$$

Let us now compute the functional derivative of the double occupancies:

$$\frac{\delta \langle \hat{n}_{a\uparrow} \hat{n}_{a\downarrow} \rangle}{\delta G_{cd}^{\downarrow\uparrow}(x_3, x_4)} \propto \delta(x - x_4)\delta_{ad}\Sigma_{dc}^{\uparrow\downarrow}(x_4, x_3) + \frac{\delta \Sigma_{aa'}^{\uparrow\sigma'}(x, y')}{\delta G_{cd}^{\downarrow\uparrow}(x_3, x_4)} G_{a'a}^{\sigma'\uparrow}(y', x), \tag{A.3}$$

where we can now easily see that the LHS does not conserve the spin along the z-axis and therefore vanishes at zero external field.

## A.2 Spin-longitudinal channel

On the other hand the expression for the irreducible vertex in the spin-longitudinal channel given in the main text is not an exact equality. Here we shall clarify where the extra approximation comes from. The irreducible vertex function in the spin-longitudinal channel reads:

$$\Gamma_{\sigma\sigma'}^{abcd}(x_1, x_2, x_3, x_4) = \frac{\delta \Sigma_{\sigma\sigma}^{ab}(x_2, x_1)}{\delta G_{\sigma'\sigma'}^{cd}(x_3, x_4)} = U_{\overline{\uparrow\downarrow}}\delta_{\sigma\bar{\sigma}'}\delta_{ab}\delta_{ac}\delta_{ad}\delta(x_1 - x_2)\delta(x_1 - x_3)\delta(x_1 - x_4)$$

$$+ U\, n_{a\bar{\sigma}}\,\delta(x_1 - x_2)\delta_{ab}\frac{\delta\lambda}{\delta G_{\sigma'\sigma'}^{cd}(x_3, x_4)}. \tag{A.4}$$

Therefore, the irreducible vertex in the spin-longitudinal channel acquires non-local and dynamical corrections, which would complicate the expression of the Bethe-Salpeter equations and further approximations are needed. In practice, one approximates the extra dynamical term to a constant deviation from the value obtained in the spin-transeverse channel, i.e. $\Gamma_{\rho/z} \sim -\Gamma_{\overline{\uparrow\downarrow}} + \delta U_{\rho/z}$.

# B Bethe-Salpeter equations

Let us define the generalized susceptibility as:

$$\chi_{1234} = \frac{\delta G(21)}{\delta h(34)}, \tag{B.1}$$

where $G(12) = -T_\tau \langle c_\alpha(x_1) c_\beta^\dagger(x_2) \rangle$ is the propagator, $x = (R, \tau)$, $1 = (\alpha, x_1)$ and $h(12)$ is the perturbing field whose action reads:

$$S_{\text{ext}} = -\int d1 d2\, h(1, 2)\bar{c}(1)c(2), \tag{B.2}$$

where in the last equations $c$ and $\bar{c}$ are Grassmann variables, and $\int d1 = \sum_\alpha \sum_R \int_0^\beta d\tau$, with $\beta = 1/k_B T$. Given the form of the external perturbation, the inverse of the non-interacting propagator reads:

$$\mathcal{G}_0^{-1}(12) = [\partial_\tau + \mu - H_0]_{12} + h(12). \tag{B.3}$$

We now want to obtain a closed equation for $\chi_{1234}$ by explicitly performing the functional derivative in Eq.(B.1). For doing so we first note that:

$$\frac{\delta G(21)}{\delta h(34)} = -\int \int d1' d2'\, G(2, 2')\frac{\delta G^{-1}(2'1')}{\delta h(34)} G(1', 1). \tag{B.4}$$

We can further develop Eq.(B.4) by making use of the Dyson equation, that reads:

$$G^{-1}(12) = \mathcal{G}_0^{-1}(12) - \Sigma(12).$$ (B.5)

In fact, by substituting Eq.(B.5) into Eq.(B.4) and using Eq.(B.3), we obtain the following identity:

$$\chi_{1234} = -G(2,3)G(4,1) + \int \prod_{i=1}^{4} di' \, G(2,2')G(1',1)\Gamma_{1'2'3'4'}\chi_{4'3'34},$$ (B.6)

where we defined the two-particle irreducible (2PI) vertex function $\Gamma_{1234} = \frac{\delta\Sigma(2,1)}{\delta G(3,4)}$. Let us express the last equation in Fourier space. For this purpose let us expand the propagators and vertices in terms of their Fourier components, i.e.:

$$f_{1234} = \frac{1}{(V\beta)^3} \sum_{kk'q} e^{i[kx_1-(k+q)x_2+(k'+q)x_3-k'x_4]} f_{\gamma\delta}^{\alpha\beta}(k,k',q),$$

$$G(1,2) = \frac{1}{V\beta} \sum_{k} e^{-ik(x_1-x_2)} G_k^{\alpha\beta}.$$ (B.7)

We first note that:

$$-G(2,3)G(4,1) = \frac{1}{(V\beta)^3} \sum_{kk'q} e^{i[kx_1-(k+q)x_2+(k'+q)x_3-k'x_4]} \chi_{0,\gamma\delta}^{\alpha\beta}(k,k',q),$$ (B.8)

where we defined the bubble terms as:

$$\chi_{0,\gamma\delta}^{\alpha\beta}(k,k',q) = -(V\beta)\,\delta_{kk'}\,G_k^{\delta\alpha}\,G_{k+q}^{\beta\gamma}.$$ (B.9)

The final equation in Fourier space reads:

$$\chi_{\gamma\delta}^{\alpha\beta}(kk'q) = \chi_{0,\gamma\delta}^{\alpha\beta}(k,k',q) - \frac{1}{(V\beta)^2} \sum_{k_1k_2} \sum_{\alpha'\beta'\gamma'\delta'} \chi_{0,\beta'\alpha'}^{\alpha\,\beta}(k,k_1,q)\Gamma_{\gamma'\delta'}^{\alpha'\beta'}(k_1,k_2,q)\chi_{\gamma\,\delta}^{\delta'\gamma'}(k_2,k',q).$$ (B.10)

## C  Improved one-loop self-energy

Let us note that from its definition the generalised susceptiblity is related to the two-particle Green's function in the following way:

$$\chi_{\gamma\delta}^{\alpha\beta}(x_1,x_2,x_3,x_4) = G_{\;\;\gamma\delta}^{(2)\alpha\beta}(x_1,x_2,x_3,x_4) - G^{\beta\alpha}(x_2,x_1)G^{\delta\gamma}(x_4,x_3).$$

Hence, we can rewrite the RHS of Eq.(3) in the following way:

$$\frac{1}{V\beta} \sum_{k\gamma} e^{-ik(x-y)} U_{\alpha\gamma} n_\gamma G_k^{\alpha\beta} + \frac{1}{(\beta V)^3} \sum_{kk'q} \sum_{\gamma} U_{\alpha\gamma} e^{-ik(x-y)} \chi_{\gamma\gamma}^{\alpha\beta}(kk'q).$$

If we substitute Eq.(B.10) into the second term of last equation we obtain the following expression:

$$-\frac{1}{(V\beta)^2} \sum_{kk'q} \sum_{\gamma} e^{ik(x-y)} U_{\alpha\gamma} G_k^{\gamma\alpha} G_{k+q}^{\beta\gamma} + \frac{1}{(V\beta)^4} \sum_{kk'qk_1} \sum_{\gamma\alpha'\beta'\gamma'\delta'} U_{\alpha\gamma} G_k^{\alpha'\alpha} G_{k+q}^{\beta\beta'} \Gamma_{\gamma'\delta'}^{\alpha'\beta'}(kk_1q)\chi_{\gamma\gamma}^{\gamma'\beta'}(k_1k'q),$$

(C.1)

Table 1: Relation between indices expressed in the compact and extended notations.

| | |
|---|---|
| $\alpha$ | $(a, \sigma)$ |
| $\beta$ | $(b, \sigma')$ |
| $\gamma$ | $(c, \sigma'')$ |
| $\alpha'$ | $(a_1, \sigma_1)$ |
| $\beta'$ | $(a_2, \sigma_2)$ |
| $\gamma'$ | $(a_3, \sigma_3)$ |
| $\delta'$ | $(a_4, \sigma_4)$ |

which is a generic and exact expression of the RHS of Eq.(3). Now we shall specialize to the antiferromagnetic phase of the Hubbard model, and approximate the vertex function to a local quantity that does not depend on the crystalline momenta. In order to do so it is useful to explicitly express the spin-orbital indices in sub-lattice and spin indices as shown in Table 1.

Furthermore, if we assume spin-conservation we can express the irreducible vertex function as follows:

$$\Gamma^{a_1 a_2}_{a_3 a_4}|^{\sigma_1 \sigma_2}_{\sigma_3 \sigma_4} \sim \delta_{a_1 a_2} \delta_{a_1 a_3} \delta_{a_1 a_4} (\Gamma^{a_1}_{\sigma_1 \sigma_2} \delta_{\sigma_1 \sigma_2} \delta_{\sigma_3 \sigma_4} + \Gamma^{a_1}_{\sigma_1 \bar{\sigma}_1} \delta_{\sigma_1 \bar{\sigma}_2} \delta_{\sigma_3 \bar{\sigma}_4} \delta_{\sigma_1 \sigma_3}), \qquad (C.2)$$

where we used the following notation $\Gamma^a_{\sigma \sigma'} = \Gamma^{aa}_{aa}|^{\sigma \sigma}_{\sigma' \sigma'}$ and $\Gamma^a_{\sigma \bar{\sigma}} = \Gamma^{aa}_{aa}|^{\sigma \bar{\sigma}}_{\bar{\sigma} \sigma}$. Substituting Eq.(C.2) into Eq.(C.1) we obtain the following expression for the equation of motion in momentum space:

$$\Sigma^{ab}_{\sigma}(k) - U n_{a\bar{\sigma}} = \frac{U}{(V\beta)^3} \sum_{k_1 k' q \sigma_1} G^{ab}_{\sigma}(k+q) \Gamma^b_{\sigma \sigma_1}(\nu \nu' \omega) \chi^{ba}_{\sigma_1 \bar{\sigma}}(k_1 k' q). \qquad (C.3)$$

We notice that in this representation the self-energy is expressed in terms of the longitudinal scattering channel only. It is possible to obtain an equivalent expression where the transverse vertex and susceptibility appear by using the following crossing relation:

$$G^{(2)\beta\alpha}_{\gamma\gamma}(y, x+0^-, x+0^+, x) = -G^{(2)\beta\gamma}_{\gamma\alpha}(y, x, x+0^+, x+0^-). \qquad (C.4)$$

Plugging the last equation into the equation of motion in Eq.(3) and following similar passages to the ones we did for obtaining Eq.(C.3), we obtain the following expression for the self-energy:

$$\Sigma^{ab}_{\sigma}(k) - U n_{a\bar{\sigma}} = -\frac{U}{(V\beta)^3} \sum_{k_1 k' q} G^{ab}_{\bar{\sigma}}(k+q) \Gamma^a_{\sigma \bar{\sigma}}(\nu \nu' \omega) \chi^{ab}_{\sigma \bar{\sigma}}(k_1 k' q). \qquad (C.5)$$

In TPSC the irreducible vertices are local and static, i.e. they do not depend on the Mastubara frequencies and further simplification arise. In particular, if we assume static and local vertex functions, if we average Eqs.(C.3,C.5) we obtain the following expression for the one-loop improved self-energy:

$$\Sigma^{ab}_{\sigma}(k) - U n_{a\bar{\sigma}} = -\frac{U}{2V\beta} \sum_{q} G^{ab}_{\bar{\sigma}}(k+q) \Gamma^a_{\sigma \bar{\sigma}} \chi^{ab}_{\sigma \bar{\sigma}}(q) + \frac{U}{2V\beta} \sum_{q \sigma_1} G^{ab}_{\sigma}(k+q) \Gamma^a_{\sigma \sigma_1} \chi^{ab}_{\sigma_1 \bar{\sigma}}(q).$$

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
