# Peer review of "Two-particle self-consistent approach for broken symmetry phases"

_SciPost Physics, doi:SciPost Phys. 18, 077 (2025)_

## Round 1 · Referee Report · Anonymous (Referee 1) · 2024-8-11

Strengths

Interesting method development.

Weaknesses

The manuscript is clearly addressed to a specialist audience. This is also reflected by a selective choice of references.

Report

The present work develops a two-particle self-consistent (TPSC) approach for broken symmetry phases and applies it to the three-dimensional Hubbard model. A benchmark against the Quantum Monte Carlo (QMC) data of Ref. [13] looks promising.

However, the manuscript clearly addresses a specialist audience. For example, there is a big leap from Eq. (1) to Eq. (2), i.e., as of the second page of the text, the author is talking to specialists only. Moving parts of Appendix A into the main text may help, but I expect that the manuscript will remain very technical nevertheless.

Another point if the exponent $\beta=1/2$ mentioned at the beginning of section 4. For the three-dimensional Heisenberg ($O(3)$) universality class, it should rather be $\beta \approx 0.369$, see for example M. Campostrini et al., Phys. Rev. B 63, 214503 (2001) (also mentioned in Ref. [13]). To me, this is a strong indication that TPSC remains a mean-field theory after all, and I would find a related discussion plus references appropriate.

Furthermore, there are other (lattice) QMC investigation of the Néel transition in the half-filled three-dimensional Hubbard model. P. R. C. Kent et al., Phys. Rev. B 72, 060411(R) (2005) and S. Fuchs et al., Phys. Rev. Lett. 106, 030401 (2011) may be a good starting point to access further literature (actually, both references are also cited in Ref. [13]).

There are a number of further relatively minor points that I mention in "Requested changes".

To conclude, I believe that there is some interest in this work such that it ultimately merits publication in some form. However, my impression at least of the present manuscript is a minor technical progress that is of interest mainly to specialists. Therefore, I recommend transfer of a suitably revised version to SciPost Physics Core.

Requested changes

1- Move part of Appendix A to the beginning of section 3 in order to make the discussion more self-contained. 2- Add discussion to the result $\beta=1/2$ at the beginning of section 4 and cite relevant references, such as Phys. Rev. B 63, 214503 (2001) . 3- Mention further investigations of the half-filled three-dimensional Hubbard model such as P. R. C. Kent et al., Phys. Rev. B 72, 060411(R) (2005) and S. Fuchs et al., Phys. Rev. Lett. 106, 030401 (2011) and compare, e.g., the Néel temperature $T_N$ with these. 4- I liked the last three sentences of the Introduction (section 1) and hope that I have passed the test. 5- I believe that many equations would fit on one line, which would render the manuscript more readable. Examples: Eqs. (4), (10), (16), (18), (19), (21), (28), (37), (40), and (42). Same for a reduction of Eqs. (29) and (41) from three to two lines. 6- Between Eqs. (5) and (6) there is an abbreviation "BSE" that has not been introduced. 7- Appendix C is quite short such that the cross-references generate unnecessary overhead. Why not move the content of the appendix to the appropriate place in section 4? 8- In panels (c) and (d) of Fig. 3, clarity would be improved of the labels "TPSC" and "RPA" had the same colors as the corresponding lines. 9- I can guess what the bar means, e.g., in the $\delta_{a\bar{b}}$ below Eq. (17), but I believe that an explanation would be helpful. 10- If the meaning of the bar in the $\overline{y}$ in Eq. (18) and below was specified, I have missed this description. 11- Below Eq. (41), there is a reference to "The last equation", but Eq. (41) actually is not an equation. 12- Some preprint references are actually published. Maybe some appeared only after submission of the manuscript, but I still recommend an update. Specifically: [13] is published in Phys. Rev. Lett. 132, 246505 (2024). [19] is published in Phys. Rev. B 109, 045155 (2024). [28] is published in Phys. Rev. B 108, 075144 (2023). [34] is published in Phys. Rev. B 109, 075143 (2024). 13- The URLs in Refs. [29,30] are redundant and could be omitted. 14-There are issues with the English text, such as: a) The manuscript mixes British and American English. For example, there are occurrences of "magnetisation" (British English) and "magnetization" (American English). I recommend that the author settles on one version and runs the manuscript through an appropriate spellchecker. b) There are several instances of "as following" which in my opinion should read "as follows". c) There is a duplicate "for for" in the caption of Fig. 3. Overall, I believe that the manuscript would benefit from careful proofreading, preferably also from somebody else than the author, and ideally a native English speaker.

Recommendation

Accept in alternative Journal (see Report)

  • validity: high
  • significance: ok
  • originality: ok
  • clarity: good
  • formatting: good
  • grammar: reasonable

Author:  Lorenzo Del Re  on 2024-09-11  [id 4763]

(in reply to Report 1 on 2024-08-11)

We have attached a detailed response to the referee’s comments in PDF format, along with a diff file that highlights the changes between the previous and revised versions of the manuscript.

Attachment:

reply_and_diff_oCTE1bl.pdf

---

## Round 2 · Referee Report · Anonymous (Referee 2) · 2024-10-15

Report
In the work by Lorenzo Del Re entitled “Two-particle self-consistent approach for broken symmetry phases,” the author introduces a generalization of the Two-Particle Self-Consistent (TPSC) method to treat broken SU(2) magnetic phases of the Hubbard model. The method is applied to the antiferromagnetic phase on a cubic lattice, allowing the author to observe a Higgs mode below the quasi-particle continuum, which is not visible in mean-field (RPA) calculations. The manuscript is well-written and contains a detailed derivation of the method, with all approximations explicitly discussed.
The topic of the work is timely. Indeed, there are not many methods that enable the calculation of correlated systems within broken symmetry phases, and this research direction is far from being complete. In this regard, I kindly disagree with the previous referee's statement that it “is clearly addressed to a specialist audience,” especially since the formation of various dynamically symmetry-broken phases is one of the hallmarks of correlated systems. In addition, the non-symmetry-broken version of the TPSC method has already been successfully applied to realistic systems (see, e.g., [Phys. Rev. Lett. 123, 256401 (2019)]), so the broken-symmetry formulation represents an important extension of the method.
Nevertheless, there are a few important drawbacks that should be addressed before this work can be published.
1) In the abstract, the author asserts that the developed method is "efficient yet reliable." However, in my view, the manuscript does not contain enough information to conclusively evaluate the reliability of the method, particularly regarding the limits of its applicability. For instance, a recent study [arXiv:2410.00962] suggests that the non-symmetry-broken TPSC approach fails even at moderate interaction strengths. If I am not mistaken, the claim regarding the method's reliability is based solely on a comparison of the magnetization and double occupancy with exact results from the diagrammatic Monte Carlo (DiagMC) method. While this comparison is indeed impressively accurate, the calculations are performed very deep inside the ordered phase, where electronic correlations are expected to be significantly diminished. Moreover, magnetization and double occupancy are local and static quantities, which are among the "easiest" to compute accurately. Therefore, it would be beneficial if the author could provide a comparison of these quantities at higher temperatures, closer to the transition temperature. Additionally, it would be valuable to see a comparison of momentum- and frequency-dependent quantities, such as the self-energy, across different temperature and interaction regimes. The self-energy could be taken from the same DiagMC calculations used for magnetization and double occupancy. Alternatively, these quantities could be compared with results from DMFT and the Dynamical Vertex Approximation (D$\Gamma$A). In fact, the author of this manuscript is also the first author of the work [Phys. Rev. B 104, 085120 (2021)], which introduces the D$\Gamma$A method for spontaneously broken SU(2) symmetry.
2) In my opinion, one of the key advantages of the TPSC method is its ability to account for the momentum- and frequency-dependent self-energy. This feature enables calculations for realistic materials [Phys. Rev. Lett. 123, 256401 (2019)] and allows the method to be combined with DMFT (see, e.g., [Phys. Rev. B 107, 235101 (2023)]) to non-perturbatively account for the effects of local correlations. However, although the "improved" (momentum- and frequency-dependent) self-energy is introduced in Eq. (21), it seems that only the local self-energy given by Eq. (8) is used in the actual numerical calculations. As a result, the method can no longer be combined with DMFT and closely resembles a simple extension of the Hartree-Fock (HF) theory. In this context, I agree with the previous referee that, in its current form, the work represents only "minor technical progress." It would be great, therefore, to see some results that involve the calculation of the "improved" self-energy.
3) One might expect that, at small values of $U$, the results of the developed TPSC method would coincide with those of the HF method. In fact, at $U = 3$, one can observe that the value of both spin vertices match the bare interaction. However, the magnetizations obtained at $U = 3$ using TPSC and HF are still quite different, and the two curves do not appear to converge at smaller values of U. Could the author comment on why the results presented in Figure 3(d) and Figure 4 for small values of U are not consistent?
4) I was somewhat confused to discover that the vertex corrections discussed throughout the manuscript are not three-frequency- and/or momentum-dependent objects, but rather scalar quantities that correspond to a renormalized bare interaction. This is likely the terminology used within the TPSC community, but it would be helpful to clarify this point somewhere in the text to avoid confusion with actual vertex functions, as used, for example, in D$\Gamma$A.
5) I am puzzled by the fact that the susceptibilities in the spin $z$ and charge density ($\rho$) channels have such a simple form. Typically, in the presence of spin polarization, the spin $z$ channel becomes intertwined with the $\rho$ channel, and the same applies to the spin $x$ and $y$ channels. Could the author comment on why, in this work, the spin $z$ and charge density $\rho$ channels can be easily decoupled, while the spin $x$ and $y$ channels are still coupled to each other?
6) Could point 5) be related to some ambiguity in formulating the symmetry-broken TPSC approach, similar to the one present in the multi-orbital formulation of the method [arXiv:2410.00962]?
7) Could the author provide the results for the spin transverse channel along with the spin longitudinal susceptibility already shown in Figure 5?
8) Could the author comment on how difficult it would be to extend the developed approach to handle more sophisticated spin-ordered phases, such as spin spirals or even more complex incommensurate orderings?
9) The discussion in the Introduction might benefit from the following citations, which could broaden the scope and interest in the current work: Phys. Rev. B 104, 085120 (2021) — D$\Gamma$A method for spontaneously broken SU(2) symmetries Phys. Rev. Lett. 123, 256401 (2019) — materials calculations using the TPSC method Phys. Rev. B 107, 235101 (2023) — TPSC+DMFT approach arXiv:2410.00962 — multi-orbital extension of the TPSC method
In addition, there has been recent development of a fluctuating field method for symmetry-broken phases, based on the variational optimization of an effective bare interaction in a given instability channel, which likely shares some similarities in spirit with the TPSC method: Phys. Rev. B 102, 224423 (2020), Phys. Rev. B 105, 035118 (2022), Phys. Rev. B 108, 035143 (2023), Phys. Rev. B 108, 205156 (2023).
Recommendation
Ask for major revision
The attached file contains responses to all referees' comments, along with a diff file highlighting the changes made between the revised and original versions.

Author: Lorenzo Del Re on 2024-11-14 [id 4960]
(in reply to Report 4 on 2024-10-26)We have attached the complete response to all referees, along with a diff file comparing the previous and updated versions of the manuscript, available on the dialog page for report #1.

---

## Round 2 · Referee Report · Nicolas Martin (Referee 3) · 2024-10-18

Strengths
1- The derivation of the TPSC equations for the broken symmetry state is done well. 2- Interesting qualitative differences between the TPSC susceptibility and RPA are presented, especially in regards to the Higgs mode.
Weaknesses
1- Parts of the derivation could be clarified 2- Implementation notes should be clarified to insure reproducibility of the results
Report
Overall, the presentation is very good, and enough details are given to allow the reader to reproduce most of the steps. The results for the three-dimensional Hubbard model do a good job of showing how the method compares with RPA and exact Monte-Carlo results.
Some minor aspects of the article could be clarified. The proposed modifications are presented in the next section. I believe that, once those minor changes are made to the article, the article should be published in SciPost Physics
Requested changes
1- In the caption of Fig. 1, it would be helpful to specify that the diagram refers only to the first-level self-energy in TPSC, and not the improved self-energy computed in Sec. 3.3. 2- The gap equation for the order parameter (Eq. 13) is presented with no explanation to how it can be derived. A bit more justification would help the reader understand the method. 3- In the footnote at the bottom of page 5, there is an extra period after “zero-field”. 4- On page 6, under Eq.10, the author writes “Since the vertex function in Eq. (8)”. I believe the correct equation to refer here is Eq. 9. 5- The method that was used by the author to compute the spatial Fourier transforms was clearly presented in Sec. 4. However, there is no mention of the way the time-frequency Fourier transforms were computed (nor is there mention of the analytic continuation method that was used to compute the real-frequency susceptibility in Fig. 5). Mentioning the methods used would help in making the work presented more easily reproducible. 6- In most of the work, the terms “spin-transverse” and “spin-longitudinal” are used to describe the scattering channels. However, in some cases (pages 6, 7 and 14 specifically), the order is reversed, with the terms “longitudinal-spin” and “transverse-spin”. It would probably be better to stick to one of the two conventions. 7- In Fig.3c, the label of the y-axis is at the bottom of the axis, in contrast with the other subplots (where the label is centered). Centering it would help readability. 8- In Fig.5c and d, the vertical dashed lines marking the start of the particle-hole continuum are hard to see. Making them a bit darker would help. 9- In the same panels, the vertical arrow used to label the curves could easily be mistaken for dirac peaks in the data. Reorienting them diagonally (or using the same legend as the one used in Figs. 3 and 4) would prevent this misunderstanding.
Recommendation
Ask for minor revision
Author: Lorenzo Del Re on 2024-11-14 [id 4961]
(in reply to Report 2 by Nicolas Martin on 2024-10-18)We have attached the complete response to all referees, along with a diff file comparing the previous and updated versions of the manuscript, available on the dialog page for report #1.

---

## Round 2 · Referee Report · Anonymous (Referee 1) · 2024-10-19

Report
However, I still believe that SciPost Physics Core would be the more appropriate venue. One of the reasons for this assessment is that the universality class of the phase transition does not come out correctly from TPSCA, i.e., the accuracy of the approach is necessarily limited, even if I am grateful for the clarification (compare beginning of section 4). However, I admit that this assessment has a subjective component; maybe I am also biased by the previous version that definitely addressed a specialist audience.
There are still a small number of minor typographic issues that I list as "Requested changes". In my opinion, these could be addressed during production.
Requested changes
1- Panels (c) and (d) of Fig. 5: It may be clearer if the direction if the arrows were inverted (compare also comment 9 by Referee 3). 2- I think that the last two lines of Eq. (25) would fit on one line, and this would improve readability of this equation. 3- Two lines below Eq. (37): "in" $\to$ "into". 4- Correct lower-casing of names in titles of References [4,12,13,15,17,18,23,24,33,44-50,52,54,66] ("Nambu", "Goldstone", "Hubbard", "Néel", "Monte Carlo", "Heisenberg", "Higgs", "Mott", "Fermi", "Hedin", "Weyl"). 5- Correct chemical formulas in titles of References: [31] "LiFeAs" [60] "RuO$_2$" [61] "$\alpha$-MnTe" [64] "KV$_2$Se$_2$O" [65] "CoNb$_4$Se$_8$"
Recommendation
Accept in alternative Journal (see Report)
We have attached the complete response to all referees, along with a diff file comparing the previous and updated versions of the manuscript, available on the dialog page for report #1.

---

## Round 2 · Referee Report · Anonymous (Referee 4) · 2024-10-26

Strengths
This a nice and well-written paper on a topic of present interest. TPSC was developed in the past as a method to describe correlated phases beyond mean field in a computationally efficient way. A hallmark of TPSC is that it includes momentum- and frequency-dependent self-energies, and has been successfully applied to understand the nature of paramagnetic phases in correlated metals, in the parameter region where the method is applicable.
The present manuscript introduces a valid extension to antiferromagnetic broken-symmetry phases that is worth publishing and, in my opinion, suitable for publication in Scipost.
Weaknesses
some of them will be also present in the broken-symmetry phases. Nevertheless, this is not a reason for not pursuing extensions of this approach, as far as one is in a reliable phase-space region, specially considering the fact that the approach is computationally feasible.
Referees 2 and 3 ask very valid points to the author, that I second.
Report
Recommendation
Ask for minor revision

---

## Round 2 · Author Response

Thank you for providing the referee reports. We are grateful to the referee for their careful, positive, and constructive feedback, which has helped us to restructure the paper and make it more accessible to a broader audience. Additionally, addressing the referee’s comments point by point allowed us to further clarify that our method extends beyond mean-field approaches and holds great potential for applications in the emerging field of altermagnetism.
Given the significant improvements we have made to fully address the referee’s requests, we believe that this revised version of our paper is now suitable for publication in SciPost Physics.
Best Regards
Lorenzo Del Re

---

## Round 2 · List of Changes



---

## Round 3 · Referee Report · Anonymous (Referee 4) · 2024-11-16

Report

The author answered satisfactorily to all open points by the referees and I recommend publication of the submitted revised manuscript

Recommendation

Publish (easily meets expectations and criteria for this Journal; among top 50%)

---

## Round 3 · Referee Report · Nicolas Martin (Referee 3) · 2024-11-29

Report

The modifications to the manuscript have addressed in a satisfactory manner all the points raised by referees. I believed the revised version should be published in SciPost Physics.

Recommendation

Publish (easily meets expectations and criteria for this Journal; among top 50%)

---

## Round 3 · Referee Report · Anonymous (Referee 2) · 2024-12-2

Report

I would like to thank the Author for considering my suggestions and for performing additional calculations in the revised version of the manuscript. I would be happy to recommend the publication of this work in SciPost Physics provided the Author addresses two remaining issues:

  1. In their reply to my first question regarding the reliability of the method, the Author mentions that there are no diagrammatic Monte Carlo data available for comparison at higher temperatures. However, the Author did not comment on the possibility of comparing the results of the developed TPSC method to those obtained using the dynamical vertex approximation or other method. Therefore, as I pointed out in the previous review round, the manuscript still does not provide sufficient information to conclusively evaluate the reliability of the method, as claimed in the Abstract. In this regard, the claim of reliability should be removed from the Abstract unless the Author provides a clear and sufficient justification.

  2. Regarding my previous question (5), I am surprised to read that the Author has limited their theoretical method development to a half-filled bipartite lattice, which is rarely realized in actual materials. While I understand that this particular case significantly simplifies the implementation of the method and the numerical calculations, I strongly believe that the Author should at least derive the method for a general case and present the corresponding equations without relying on the particle-hole symmetry specific to the half-filled bipartite case.

Recommendation

Ask for minor revision

  • validity: -
  • significance: -
  • originality: -
  • clarity: -
  • formatting: -
  • grammar: -

Author:  Lorenzo Del Re  on 2025-01-17  [id 5132]

(in reply to Report 3 on 2024-12-02)

A detailed reply and a diff file have been added as a file attachment in pdf format.

Attachment:

reply_and_diff_v3_v4_compressed.pdf

---

## Round 3 · Author Response

Dear Editor,

Thank you for inviting new referees and providing their reports. We appreciate the referees' careful, positive, and constructive feedback, which has helped us better highlight the potential (and limitations) of the proposed method.

We have provided point-by-point responses to all referees' comments and requests, and we have added the additional data requested by Referee 2.

Given the substantial improvements made to address all comments and requests, we believe the revised version of our paper is now suitable for publication in SciPost Physics.

---

## Round 3 · List of Changes

We have attached the complete response to all referees, along with a diff file comparing the previous and updated versions of the manuscript, available on the dialog page for report #1.

---

## Round 4 · Author Response

We appreciate the positive feedback from all Referees and are pleased that they have recommended our paper for publication. In this resubmission, we have addressed the minor revisions requested by Referee 2.
Thank you for considering our work.

---

## Round 4 · List of Changes



---

## Editorial Decision

published